# Nanoscopic Approach to Study the Early Stages of Epithelial to Mesenchymal Transition (EMT) of Human Retinal Pigment Epithelial (RPE) Cells In Vitro

**DOI:** 10.3390/life10080128

**Published:** 2020-07-30

**Authors:** Lilia A. Chtcheglova, Andreas Ohlmann, Danila Boytsov, Peter Hinterdorfer, Siegfried G. Priglinger, Claudia S. Priglinger

**Affiliations:** 1Institute of Biophysics, Johannes Kepler University (JKU) Linz, Gruberstrasse 40, 4020 Linz, Austria; danila.boytsov@jku.at (D.B.); Peter.Hinterdorfer@jku.at (P.H.); 2Department of Ophthalmology, Munich University Hospital, Ludwig-Maximilians-University (LMU) Munich, Mathildenstrasse 8, 80336 Munich, Germany; andreas.ohlmann@med.uni-muenchen.de (A.O.); Siegfried.Priglinger@med.uni-muenchen.de (S.G.P.); Claudia.Priglinger@med.uni-muenchen.de (C.S.P.)

**Keywords:** human retinal pigment epithelial (hRPE) cells, type 2 epithelial to mesenchymal transition (EMT), atomic force microscopy (AFM), F-actin cytoskeleton, microvilli, geodomes

## Abstract

The maintenance of visual function is supported by the proper functioning of the retinal pigment epithelium (RPE), representing a mosaic of polarized cuboidal postmitotic cells. Damage factors such as inflammation, aging, or injury can initiate the migration and proliferation of RPE cells, whereas they undergo a pseudo-metastatic transformation or an epithelial to mesenchymal transition (EMT) from cuboidal epithelioid into fibroblast-like or macrophage-like cells. This process is recognized as a key feature in several severe ocular pathologies, and is mimicked by placing RPE cells in culture, which provides a reasonable and well-characterized in vitro model for a type 2 EMT. The most obvious characteristic of EMT is the cell phenotype switching, accompanied by the cytoskeletal reorganization with changes in size, shape, and geometry. Atomic force microscopy (AFM) has the salient ability to label-free explore these characteristics. Based on our AFM results supported by the genetic analysis of specific RPE differentiation markers, we elucidate a scheme for gradual transformation from the cobblestone to fibroblast-like phenotype. Structural changes in the actin cytoskeletal reorganization at the early stages of EMT lead to the development of characteristic geodomes, a finding that may reflect an increased propensity of RPE cells to undergo further EMT and thus become of diagnostic significance.

## 1. Introduction

The retina represents several tissues that reside at the back of the eye. It consists of two main layers: an inner neuroretinal layer, which is composed of photoreceptors and cells that process the outputs from the photoreceptor cells and send them to the brain; and an outer pigmented layer termed the retinal pigment epithelium (RPE). During early organogenesis, the RPE is derived from the neural ectoderm of the optic vesicle as a partner of the neural retina. After maturation, the neural retina works as a complex neural circuit encoding visual information and sending it to the brain through the optic nerve, while the RPE performs an amazing variety of activities indispensable for the proper function and survival of the photoreceptors [1] (selective blood-retinal barrier; absorption of stray light; retinoid processing and recycling; nutrient transport, metabolites, and metabolite clearance; photoreceptor renewal), making it responsible for the maintenance of visual function. 

In a healthy adult eye, the RPE is a dense monolayer of highly polarized pigmented cuboidal epithelial cells located between the neural retina and the choriocapillaris of the choroid (Figure 1).

Healthy RPE cells are in a post-mitotic state, but they retain the ability to divide and do so in response to environmental stimuli. For example, RPE proliferates rapidly in response to injury. In mild injuries involving only the photoreceptors and the RPE (e.g., in the case of laser photocoagulation), proliferating RPE cells quickly re-establish a continuous RPE monolayer. However, following retinal insults ranging from inflammation or complicated cases of retinal detachment to aging, RPE cells go in their “repair mode”: they detach from Bruch’s membrane and start to both proliferate and migrate while undergoing epithelial-to-mesenchymal transition (EMT) (or cell-type switching) from their differentiated phenotype into fibroblast-like (also called myofibroblast [5]) or macrophage-like cells [6,7], but are not effective in creating a new functional retina-RPE complex. In clinical practice, fibroblast-like RPE cells were found to be involved in the formation of subretinal (choroidal neovascular) membranes in the “wet” form of age-related macular degeneration (AMD) [8]. However, the more clinically relevant event is the migration of dedifferentiated RPE cells into the vitreous cavity and consequent development of elastic scar tissue sheets termed epiretinal membranes in a clinical condition referred to as proliferative vitreoretinopathy (PVR). Both phenotypes of dedifferentiated RPE cells largely contribute to the formation of epiretinal membranes [6,9,10,11], which contract and distort the retina in such pathologies as PVR [6,12,13], leading to repetitive retinal detachment with severe and persistent visual loss. In early PVR, ophthalmologists can discern the pigmented cells in the vitreous cavity, but from a clinical point of view it is not possible to quote whether the cells are resting or about to dedifferentiate. The treatment of PVR is primarily surgical, but surgery is not performed until the advanced stages of PVR when the membranes start to distort the retina. Once established, PVR is difficult to cure by surgical means alone and, despite the incessant advances in vitreoretinal surgery, PVR is the main reason for treatment failure after primarily successful retinal detachment surgery.

There is no universally accepted term for what occurs with RPE cells in vivo in injury-related activities and under pathological conditions. In the literature, such phenotypic changes in RPE cells are referred to as metaplasia [10]; dedifferentiation [7]; transdifferentiation [8,14]; EMT [9,15]; or, more precisely, type 2 EMT [16], a non-malignant EMT associated with wound healing and tissue repair responses [17]. It should be noted that the RPE cells do not become macrophages and fibroblasts—they just adopt their appearance and operate like them—but in certain laboratory conditions the cells can rapidly revert to an epithelioid form and thus undergo a reverse mesenchymal to epithelial transition (MET) [6,18,19].

RPE cells are recognized as the key players in numerous retinal pathologies; undoubtedly, the search for molecular cues regulating their dedifferentiation and the ways of reversing this process is obviously of both fundamental and practical interest with special regard in RPE transplantation and the effective treatment of retinal diseases such as PVR. A series of research approaches to control the dedifferentiation of RPE cells have already taken shape [20,21]. Most of them are studies on signaling pathways that revealed novel molecular targets for suppressing the dedifferentiation process of RPE cells. Previous clinical trials assessing the efficacy of anti-proliferative and anti-inflammatory substances have yielded mixed results [22,23]. Thus, no safe or sufficiently effective pharmacological agent has been established in the clinical routine so far. Even more importantly, there is a need for a reliable diagnostic tool to predict PVR in order to justify an early surgical intervention to prevent progression to the advanced stages of the disease, which are consequently hard to cure. In the past, most efforts in this respect focused on cytokine levels or the identification of single nucleotide polymorphisms [24,25,26]. At present, there are no reports that take the phenotypic and biochemical changes in RPE cells into consideration as factors of diagnostic or prognostic value, although the analysis of cellular infiltrates of the vitreous is routinely used for the differential diagnosis of ocular pathology, such as intraocular lymphoma.

The current knowledge on EMT of RPE cells is primarily based on optical microscopic (immunohistochemistry) and electron microscopic (EM) findings of human specimen of epiretinal membranes or cells isolated from the vitreous of animal models for pathologies such as PVR, but not on the direct analysis of the cells found to be dispersed in the vitreous of humans [9,27,28,29]. One reason for the lack of such data might be that studies on vitreous samples using optical techniques are hampered by the low number of cells found in the vitreous. Furthermore, the intense pigmentation of RPE cells imposes technical limitations for routine light microscopic staining techniques or immunofluorescence. As an alternative, a non-optical microscopic surface technique such as atomic force microscopy (AFM), which was developed by Binnig and co-workers in 1986 [30], represents a versatile platform for a high-resolution structural (several nm) and picoNewton (1 pN = 10^−12^ N) sensitive force analysis of single cells to overcome these limitations [31,32,33,34,35,36]. Moreover, the AFM has several particular advantages over EM, including the possibility of examining biological processes under/or near physiological conditions (e.g., in the liquid environment) with marginal sample preparation (no labelling or coating). AFM has become a perfect tool to evaluate the structure and function of cellular membranes in detail. The use of AFM in cell biology is thus in continuous exponential growth [37]. Nevertheless, AFM applications in vision science unfortunately remain still very limited, and in recent times only one brief AFM analysis has been performed on dried human RPE cells [38]. 

Aspects of the RPE dedifferentiation process seen in various degenerative or proliferative vitreoretinal diseases are known to be imitated when primary RPE cells are cultured on hard support surfaces (e.g., plastic petri dishes, glass slides, etc.), thus providing a well characterized in vitro model for a non-malignant type 2 EMT [7]. Like most epithelial cells that are isolated from tissues and placed in culture, the phenotype of human RPE cells in vitro is more variable than for the same cells in situ [39]. The main focus of the present work is to demonstrate the constructive and successful use of the AFM in order to gain a better understanding of the early process of the EMT of RPE in vitro. Thus, we exploit the heterogeneity of RPE cells in vitro to highlight the suitability and applicability of AFM in order to study the various initial steps of the RPE dedifferentiation that exist in vivo and thus could be differentially linked to the development of retinopathies based on their matrix-remodeling, migratory, and proliferative activities.

## 2. Materials and Methods

### 2.1. Isolation and Culture of Primary Human RPE Cells

Human retinal pigment epithelial (hRPE) cells were isolated from human cadaver eyes obtained from the Eye Bank of the Department of Ophthalmology at the Kepler University Hospital (Linz, Austria) and the Munich University Hospital Eye Bank (Munich, Germany), processed within 8 to 24 h after death. The methods for securing the human tissue were humane, included the written informed consent and approval of the relatives, and complied with the Declaration of Helsinki. The practical isolation of RPE cells from human cadaver eyes for this study and the respective consent procedure was approved by the local ethics committee. hRPE cells were harvested from 12 post-mortem eyes (the average donor age was 50 ± 9 years) without ophthalmological pathologies following the well-established approved routine [5,40]. The suspended cells were then plated on glass slides (with diameter of 30 mm) in a 35 mm diameter petri dish and grown in Dulbecco’s modified Eagles medium (DMEM, Gibco) supplemented with 10% fetal bovine serum (FBS, Gibco) in a humid (95%) 5% CO_2_ atmosphere at 37 °C. As in previous studies [40], the epithelial origin of the cultivated RPE cells was fully validated and the cell samples were found to be free from contaminating macrophages and endothelial cells. About 4–10 days later, the hRPE cells reached the confluent state. Next, they were removed from the dish with a trypsin EDTA mixture and further sub-cultured. hRPE cells from passages 0 to 6 were used to study early steps of the EMT of RPE in vitro.

### 2.2. Atomic Force Microscopy (AFM)

For AFM measurements, the cells were grown until at a desired confluence state (single cells or completely confluent) and subsequently gently fixed with glutaraldehyde (final concentration 0.5%), as described in [31,41]. The AFM experiments were conducted on a 6000 ILM AFM setup (Agilent Technologies Inc., Tempe, AZ, USA) mounted on an inverted light microscope (ILM) Zeiss Observer.D1 (Zeiss, Jena, Germany) schematically represented in Figure 2A. Such a combined instrument offers the possibility of using optical images to accurately perform further AFM studies. Firstly, optical (phase contrast, PhC, or differential interference contract (DIC)) images were acquired with a 10× (air immersion) objective (Carl Zeiss Microscopy GmbH, Jena, Germany) and captured with a Hamamatsu digital camera ORCA-Flash 2.8. This procedure enables the visualization of an AFM cantilever and RPE cells at the same time (Figure 2C). Then, the AFM cantilever was positioned on a cell of interest and topographical (height and deflection) images of hRPE cells were finally collected in contact mode using a silicon nitride cantilever with nominal spring constant of 10 pN/nm (MLCT-AUHW, Bruker, CA, USA). The AFM data analysis was performed using a free open source software Gwyddion (Version 2.55, gwyddion.net), which allowed us to visualize and analyze the AFM images, and to estimate the roughness value using included procedures. The roughness values were calculated on different places on the cell surface with the fixed size of 8 × 8 µm^2^.

### 2.3. Laser-Scanning Confocal Microscopy (LSCM)

To perform correlative immunofluorescence studies, the hRPE cells were grown on glass-bottom Petri dishes (MatTek, Ashland, MA, USA) and fixed for 20 min using a 4% formaldehyde solution at room temperature. The cells were then washed with PBS buffer three times and quenched with 50 mM of NH_4_Cl for 20–30 min followed by washing with the buffer solution. To label F-actin, the cells were incubated in PBS solution containing rhodamine-phalloidin dye (1:200) (Life Technologies/Invitrogen, Carlsbad, CA, USA) for 30 min at room temperature. The excess of dye was removed by rinsing the sample with PBS buffer. The samples with macrophage-like RPE cells were additionally labelled with concanavalin A (Con A) conjugated with fluorescein isothiocyanate (FITC) dye. The cells were imaged in PBS with Fluoview FV10i (Olympus, Shinjuku, Tokyo, Japan) using a ×63, 1.2 NA water immersion objective or with a commercial laser scanning microscope (LSM 510) (Carl Zeiss Microscopy GmbH, Jena, Germany) using a ×40, 1.2 NA water immersion objective. All the confocal images of dual-labelled samples were taken as single track to avoid crosstalk. The immunofluorescence images of basal and cortical F-actin cytoskeleton were obtained from projections of corresponding Z-stack images. The optical (bright field and fluorescence) images were then processed using Fiji software [42] with adjustments made only to the brightness and contrast. 

### 2.4. RNA Isolation, cDNA Synthesis, and Real-Time Quantitative Reverse Transcription Polymerase Chain Reaction (qRT-PCR) Analyses

Total RNA from the hRPE cells was isolated with peqGOLD TriFast (Peqlab, Radnor, PA, USA) according to the manufacturer’s instructions. A photometer (Eppendorf, Hamburg, Germany) was used to measure the RNA concentration and the OD260/OD280 ratio. Only total RNA with a 260/280 ratio between 1.6 and 2.0 was used for first-strand cDNA synthesis with the iScript cDNA Synthesis Kit (Bio-Rad, Hercules, CA, USA). Real-time RT-PCR analyses were performed on a CFX Real-Time PCR Detection System (Bio-Rad). iTaq Universal SYBR Green Supermix (Bio-Rad) was used for the PCR according to the manufacturer’s protocol. The following temperature profile was used: 40 cycles with 10 s denaturation at 95 °C, 30 s of annealing and extension at 60 °C. RNA which had not been reverse-transcribed served as a negative control for the real-time RT-PCR. All the PCR primers (Table 1) were designed to span exon-intron boundaries and were purchased from Life Technologies. For relative quantification, the housekeeping gene GNB2L1 was used. The results were analyzed using the CFX optical system software (Version 2.1). The expression levels were normalized to the mRNA level of hRPE cells from p0 and p1. 

## 3. Results

### 3.1. Diversity of Different RPE Phenotypes at Confluency in Vitro: Assesment with Optical Microscopy

Within a few days in culture, the suspended (isolated) single mature differentiated hRPE cells undergo significant morphological changes (Figure 2C–E). From simple cuboids in RPE tissue, they become rounded, attached, and spread on the support surface by forming pronounced cellular protrusions such as lamellipodia (Figure 2C–E), retract microvilli (Figure 2D–E), and lose pigment (melanin granules or melanosomes) by expelling them into the culture media (Appendix A). Subsequently, the spread cells start to divide, forming small growing colonies, and the dedifferentiation process is thus initiated. 

In a confluent layer of cultured hRPE cells at early passages (p0 and p1), a mixture of different phenotypes can be detected (Figure 3A–G). Most RPE cells represent large colonies of flat epithelioid cells with different sizes and shapes. Depending on the preparation, tightly packed cells with phase-bright borders that exhibit a distinctive morphology referred to as cobblestone mosaic can be observed interspersed between flat epithelioids (Figure 3B). On one occasion, we were able to culture a complete cobblestone carpet of RPE cells (Figure 3D), which indeed most closely resemble the mosaic of healthy, mature RPE *cells in situ* [43]. Moreover, we also observed that some RPE cells were not proliferating but were rather taking up the melanosomes expelled by dying RPE cells and hereby becoming very large in size and intensively pigmented cells that can be clearly identified in optical DIC images (Figure 3F′,G), suggesting a macrophage-like phenotype of these RPE cells (flashed by yellow arrows in Figure 3A,C. At passage 1, some small colonies of RPE cells with an elongated cell shape appearance (or with a fusiform morphology) and elongated nuclei can be detected (Figure 3G), thus mimicking fibroblastic morphology, and are further referred to as a fibroblast-like phenotype.

Distinct RPE morphologies optically observed at early passages (p0 and p1) are also distinctly recognized in further subcultures (p2 and p4) (Figure 4), especially cell progeny able to retain the phenotype of the parent culture [44]. At p2, the optical findings illustrate the predominant presence of cells with a distinct flat epithelioid (Figure 4A) and/or the spindle-shaped fibroblast-like (Figure 4B) phenotypes. Some RPE cells with the flat epithelioid phenotype from p2 (Figure 4A) have a typical polygonal shape but appear bigger in size compared to the flat epithelioid cells at p0 or cobblestone patterns. Some isolated macrophage-like RPE cells are still present in the culture at passage 2 (indicated by the arrow in Figure 4B). Notably, the heterogeneity in cell size is increased with the passage number, and most of the cells become significantly enlarged (Figure 3 and Figure 4). Starting from p2, the hRPE cells at a certain cell density can undergo a spontaneous elongation, and the general form of the confluent monolayer becomes very similar to a fibroblastic appearance with a swirl pattern of tightly packed elongated cells (Figure 4B). These cells are designated as fibroblast-like RPE cells.

### 3.2. Cytoskeletal Changes in Cultured hRPE Cells: Correlative AFM and Fluorescence Structural Analysis

To characterize the different phenotypes of cultured hRPE cells that may reflect the early and intermediate stages of the EMT in vitro, we evaluated the size, shape, and cell topography of individual RPE cells and subsequently analyzed the organization of the cortical cytoskeleton together with cell membrane structures such as ruffles, protrusions, and microvilli by atomic force microscopy (AFM). The cell geometry was estimated from AFM and phase-contrast optical images, whereas correlative immunofluorescence studies were performed to visualize the F-actin filament network at the basal and apical cell surface, because actin as a major dynamic cytoskeletal protein governs the shape of a cell.

#### 3.2.1. Re-Morphogenesis of RPE Cells at Early Stages of EMT: Initial Passages 0 and p1

From our numerous observations, only primary hRPE cells (p0) in culture are able to achieve a natural honeycomb appearance of RPE cells in situ by forming cobblestone clusters (Figure 3B and Figure 5A). During an extended cultivation time of several weeks, flat epithelioids can further form a regular packing structure of polygonal cells (Figure 3C and Figure 5B). The RPE cells in the cobblestone formation are characterized by a honeycomb appearance with characteristic corrugations or ruffles, which most probably reflect retracted microvilli. These ruffles cover the entire apical cell surface (Figure 5A,B). RPE cells in cobblestone clusters can reach up to 3 µm (estimated from AFM cross-section analysis) in height and have a varying size (or longest axis) from ~7 to 15 µm (11.7 ± 3.0 µm, n = 24). In our present studies, we had only one chance to detect RPE cells with a formative cobblestone morphology; the RPE cells in this unique sample were fixed with glutaraldehyde and further evaluated only with AFM. The fluorescent measurements were unfortunately hampered by autofluorescence of glutaraldehyde molecules.

In most cases, RPE cells at early passages have a tendency to organize themselves in a network of flat epithelioids with different sizes and shapes (Figure 3A,E). Many cells maintain a polygonal shape by forming a compact cellular carpet without cell overlay and empty spaces (Figure 5C,D). AFM images illustrate flattened (or smoothed) apical cell surfaces, and on some cells the presence of several irregularly distributed actin-based cellular protrusions are recognized (Figure 5C,G). Representative fluorescence images of F-actin arrangements are demonstrated in Figure 5E–H.

At the basolateral site (or at the level of the cell-cell junctions), the flat epithelioids still possess a circumferential F-actin belt or the circumferential microfilament bundles (CMB) typical for native RPE cells in situ [3,45] (Figure 5F). Remarkably, the flat epithelioids contain abundant F-actin stress fibers clearly recognized at the basal surface of individual cells (Figure 5E), whereas the native healthy hRPE cells have a very small number of stress fibers [45]. Moreover, by performing an AFM image analysis of apical cellular parts, we found that that some flat epithelioid cells exhibit a particular cortical F-actin cytoskeleton architecture, representing interconnecting microfilaments that are arranged in a triangulated irregular network beneath the plasma membrane (Figure 5D, indicated by white arrows), which are called in the literature a “geodesic dome” or “geodome” [46]. The molecular nature of cellular protrusions (ruffles (Figure 5G) and geodomes (Figure 5H)) has been also validated by F-actin staining.

As indicated above (Figure 3), hRPE cells at early passages can also appear as inhomogeneous patches or clusters of cells with other phenotypes than epithelioids (e.g., clusters of fibroblast- and macrophage-like cells) (Appendix A). The heavy pigmentation of macrophage-like RPE cells significantly influences the proper immunofluorescence studies to quantify the F-actin morphology. As expected, melanosomes were found to be strongly autofluorescent both at 540 and 488 nm excitation, interfering with the visualization of fine filamentous F-actin cytoskeletal structures at standard wavelengths used in conventional fluorescent microscopy (Appendix A). In this regard, it was highly advantageous and exceptional to explore a non-optical scanning probe technique such as AFM to evaluate the cell membrane structures at a high resolution (Appendix A). In macrophage-like RPE cells, filamentous structures beneath the cell membrane and stable protrusions can be easily detected by AFM. Moreover, membrane-bound as well as phagocyted melanosomes can be also identified. These data confirm that the usage of AFM can help to overcome experimental limitations connected with autofluorescence and consequently has an advantage in studying the cell membrane structures at a high resolution.

#### 3.2.2. Cytoskeletal Changes in RPE Cells at Intermediate Stages of EMT: Passages 2 and 4

Following consecutive cell sub-culturing, two major distinct RPE phenotypes, namely flat epithelioids and fibroblast-like cells, develop, as is clearly recognized in optical phase contrast images (Figure 4A,B). The AFM data indicate that some RPE cells with the flat epithelioid phenotype from passage 2 (Figure 6A) are still able to maintain a typical polygonal (hexagonal) shape but appear significantly larger in size than cells in cobblestone clusters or flat epithelioid cells from early passages 0 and 1. However, in contrast to the flat epithelioids from the initial passages, cells from the later ones (p2 and p4) possess significantly less cell membrane ruffles, and the filaments of the cortical cytoskeleton become visible (Figure 6A–D). Moreover, cells with cobblestone and macrophage-like appearances were absent in these single cultures and were not observed as well in other samples and in subsequent sub-cultures. Remarkably, the presence of geodesic domes on the cell cortex of flat epithelioids (p2 and p4) is systematically confirmed by AFM assessment (Figure 6A–D). In addition, among the flat epithelioid cells it is possible to detect some particular cells (optically identified as flat epithelioids) with long stress fibers running along the cell cortex (Figure 6B, upper cell in the middle of image). With further sub-culturing (p4), there is the increased number of such distinct cells with progressively acquired “mesenchymal” characteristics (Figure 6D), which subsequently group in small colonies of elongated fusiform cells (Figure 4C). The comparative fluorescence assessment of F-actin networks in flat epithelioids reveals various actin arrangements, including mostly abundant prominent stress fibers at the basal and cortical cellular sides (typical mesenchymal characteristics) (Figure 6E,F,H), sporadically circumferential rings (epithelial feature), and in occasional cases geodome networks (Figure 6E–H).

Depending on yet-unknown factors, cultured RPE cells very often dedifferentiate into a fibroblast-like (or myofibroblast [5]) phenotypes with an elongated fusiform appearance growing in a compact monolayer (Figure 4B and Figure 6I,J). Such cells can attain more than 60 µm in length and have a width varying between approximately 8 and 20 µm (Figure 6I–J). Long stress fibers running along the cell body are clearly seen in the AFM topographical images (Figure 6I–J). As previously, the molecular origin of these microfilaments was confirmed by F-actin staining with rhodamine-phalloidin. Fluorescent images illustrate similar longitudinal actin fibers at an apical part of the cells (Figure 6L).

Our observations indicate that hRPE flat epithelioids exhibiting an epithelial typical morphology already at p0 have distinct mesenchymal characteristics, such as the presence of prominent stress fibers at the basal aspects of cells (Figure 5E). By further sub-culturing, the flat epithelioids still display an epithelial morphology but progressively gain additional mesenchymal characteristics at the level of F-actin cytoskeleton organization (the disappearance of lateral circumferential rings, the occurrence of longitudinal stress actin filaments at the apical surface) (Figure 6B,D–F). We presume that flat epithelioids are likely represent different intermediate stages of EMT, whereas fusiform RPE cells are emblematic for “completed” EMT.

### 3.3. Expression Pattern of RPE Differentiation Markers in Cultured hRPE Cells: qRT-PCR Data

The increasing grades of morphological mesenchymal transformations as described above were next investigated for the mRNA expression of RPE-signature, epithelial, and mesenchymal markers (Table 2). As determined by real-time qRT-PCR, the relative mRNA expression of the cytoplasmic visual cycle enzyme RPE65 in cells with a flat epithelioid morphology of p2 and p3 remains practically at the same level when compared to the primary cultured RPE cells (p0). In contrast, in cells with a fibroblast-like phenotype (p6), no mRNA for RPE65 could be detected. Similar findings were observed for the epithelial marker E-cadherin, whose level in flat epithelioids is partly decreased in comparison with the expression level in primary RPE cells p0, whereas it dropped to the detectable minimum in fibroblast-like cells. These results strengthen the notion that cells with a fibroblast-like morphology cease RPE and epithelial markers and thus display a higher grade of EMT. In order to further assess aspects of EMT upon the culturing of hRPE cells, the expression of alpha-smooth muscle actin (α-SMA) mRNA, a widely used marker of EMT, was determined. When compared to primary RPE cells (p0), the expression of α-SMA mRNA, although yet not statistically significantly, did increase in cells with the epithelial phenotype of passage 2–3 (2.94 ± 0.74, *p* = 0.06) and at a statistically significant level in fibroblast-like cells (3.91 ± 0.85; *p* = 0.01). Altogether, these results confirm our previous assumption that cells having the flat epithelioid phenotype express both sets of markers (epithelial as well mesenchymal) and accordingly pass only partly through EMT.

### 3.4. Scheme for RPE Cytoskeletal Reorganization during EMT: from Cobblestone to Fibroblast-Like Appeareance

According to the current view based on the numerous observations ex vivo and in vitro, the EMT of RPE includes the shift from the differentiated circumferential actin ring to a linearly arranged cytoskeleton composed of numerous stress fibers. Although MVs completely disappear upon the progression of EMT, ultrastructural insights into the intermediate stages of EMT are still missing. From our present AFM observations, we propose a scheme which illustrates the gradual switch from the RPE functional microvillous architecture through geodesic dome reorganization into mesenchymal stress fiber formation (Figure 7). RPE cells (passage 0) with a cobblestone appearance are characterized by the presence of a retracted microvillous structure (Figure 5A,B and Figure 7B); the adjacent spread flatted cells have a corrugated apical cytoskeleton with rod-like structures (Figure 7C). With further sub-culturing cobblestone cells do not show, whereas cells with characteristic polygonal networks (or geodomes) become visible (Figure 5D,H, Figure 6A,C,G and Figure 7C,D). The polygonal network consisted of a regular pattern of points (the nodes) with interconnexions (the struts). Apparently, the nodes of geodomes are originated from rod-like structures. The geodomes cover either an entire apical cell surface (Figure 7D) or appear on the periphery or the edge of cells (Figure 7C). The regions with these partial polygonal networks are connected with each other via long filaments running along the cell body (Figure 7E), thus representing precursors for the development of stress fibers in flat epithelioids (Figure 7F). The final step consequently consists of the conversion of such flat epithelioids in fibroblast-like cells (Figure 7G). 

In addition to the descriptive comparison of the specific distinctive membrane-cytoskeletal structures shown in Figure 7, we performed a statistical analysis of the AFM data in terms of the distribution of its heights, or, in other words, the cell membrane roughness. This quantitative morphological parameter can effectively characterize the micrometric-size membrane features, including cytoskeletal alterations [47,48,49]. The Rrms values obtained from different cells with a specific phenotype are summarized in Table 3. They are furthermore described as a representative mechanobiological quantifier of each phenotype. We can clearly see that the cell surface became smoother with the progression of EMT. The RPE cuboidal cells characterized by the presence of a microvillous structure have the highest roughness value of about 250–330 nm. This value significantly drops to the level of ~ 100 nm as the cells became flattened but still bear ruffles and rod-like structures on their surfaces. The most impressive result of the statistical analysis reported in Table 3 is the remarkably smaller and practically invariable value of roughness (43.5 ± 6.2 nm) measured on flat epithelioids with geodomes. The values calculated on fibroblast-like cells are at the same range, however deviate more; this effect is probably due to the different sizes of the stress fibers.

## 4. Discussion

The healthy RPE cells of an adult person are in postmitotic silent state (terminally differentiated), practically without signs of cell cycle activity [50]. A fascinating fact is that their total number appears to remain constant throughout adult life [51,52,53]. They are considered to form a monolayer of identical hexagonal in shape cells. In fact, their size, shape, and geometry depend not only on their location in the eye sphere, but also alter with age [54]. Recent experimental observations in human RPE flatmounts from healthy donors of different ages illustrate that, in contrast to a uniform polygonal (hexagonal) geometry both at the fovea and near periphery in young adults, RPE cells may lose their hexagonality, become more elongated, and enlarge in size in the ninth decade [45,51]. These observed differences in the aging eye are quite similar to the pathological changes in the RPE that occur in initial stages of age-related diseases, such as age-related macula degeneration (AMD) [55]. The normally precise packing geometry of RPE becomes highly variable in AMD; cells can lose their hexagonal shape, became rounded; enlarge in size (up to 50–60 µm), as well start to exhibit stress fibers at the basal cell surfaces [56]. In the advanced dry form of AMD, the RPE cells exhibit migratory behavior; lose their critical epithelial function; and, as a consequence, transform into mesenchymal fibroblast-like cells, undergoing type 2 EMT [16]. From another side, the RPE cells with the fibroblast-like phenotype attain many molecular characteristics of the RPE cells found in pathological epiretinal membranes in PVR [9,15,40,57]. Although a pivotal role of the RPE in the pathogenesis of AMD and PVR has been postulated for several decades, the RPE morphological status at the initial states of disease development (accordingly in the early stages of EMT) is still poorly understood. Since EMT is reversible, logically these cells are considered as potential targets for novel therapies aimed at hindering disease progression and further reversing it.

To this date, the knowledge on the ultrastructure of retinal pigment epithelial cells (RPE) in situ is based on electron microscopic and histological findings exclusively. Information with respect to the spatial distribution of cell surface transmembrane proteins and the related topography is not available so far. The AFM imaging of single cells can provide novel insight into the dynamic processes, such as nanostructural topographical changes caused by growth or drug interactions [58,59]. Since one of the key events in the RPE dedifferentiation process is the cell phenotype switching, accompanied by the cytoskeletal reorganization with changes in cell size, shape, and geometry, we screened these morphological attributes in individual dedifferentiated human RPE cells in a monolayer by AFM topographical imaging at the subcellular level with a high lateral resolution. Moreover, we present here the cell roughness, which is calculated from the corresponding AFM topographical images, as a quantitative independent measure of the morphological characteristics at the nanometre scale.

Currently, the AFM imaging of living mammalian cells remains mainly limited to resolutions in the 50–100 nm range because of the soft, fragile, corrugated, and dynamic nature of the cell membrane. To improve this, we used a gentle cell fixation [41]. This distinguished fixation procedure was found critical not only in preventing the receptor’s lateral mobility but also in allowing preserving the cell morphology, the integrity of the cell membrane, as well the filamentous structure of the cortical cytoskeleton. Moreover, it radically increases the resolution of imaging of cells to the 5–10 nm range [31,41,60] and allows quick and safe sample transportation between the research institutions, as well as multiple usages of samples and their storage for a prolonged period. We showed in this work that the AFM topographical analysis can be successfully exploited to quantitatively evaluate at the nanoscale the morphological changes such as cell shape, size, and heights, and particularly distinct arrangements of the apical cytoskeleton during the RPE dedifferentiation in vitro. 

Applying these criteria, we were able to discern the light-microscopically defined “epithelioid” phenotype, as seen at an early stage of EMT (p0), into two different AFM-based phenotypes, with the “cobblestone” phenotype being covered with microvilli and the “flat-epithelioid” phenotype, which retains the planar-apical orientation with some microvilli but is not as elevated in cell height as the cobblestone phenotype. In terms of both AFM measurements and optical criteria, the “cobble-stone” phenotype most resembled the healthy, native RPE. When EMT proceeded as it does in vitro when the RPE cells are sub-cultured [39,44], the cultures became heterogenous. An assortment of the cells retained the light microscopic “epithelioid” parental phenotype and were enlarged, while AFM-measurements additionally clearly revealed the loss of microvilli and the appearance of interconnecting fibres arranged in a triangulated irregular network, which was not found in early EMT “epithelioids”. Thus, whereas the myo-fibroblastic, “fusiform” phenotype with stress fibres longitudinally spanning the cell body and the “macrophage”-phenotype may also be anticipated by light microscopy, AFM imaging enabled us to differentiate the different subtypes of cells with an epithelioid phenotype, revealing the appearance of a geodesic dome pattern on “epithelioid” RPE cells upon the progression of EMT, a finding that may reflect an increased propensity of the cells to undergo further EMT. 

In the present work, we observed not only typical arrangements of cortical cytoskeleton such as linear stress fibres and circumferential rings of microfilaments in the dedifferentiated RPE cells [61], but also revealed cytoskeletal geodesic dome assemblies consisting of highly ordered polygonal F-actin arrays in flat epithelioids at different passages. Such cytoskeletal polygonal networks or geodomes were first described by Lazarides and co-workers about 40 years ago [46,62,63,64], and were subsequently found and extensively investigated in many cultured cells [65,66,67,68,69,70,71,72,73,74]. According to the previous research, the geodesic domes were mostly observed in cultured primary cells such as rat and chick embryonic cells, chick endoderm cells, neonatal rat cardiac fibroblasts, chick cardiac myocytes, the hepatocytes of adult rats, and others. 

In ocular cells, such cytoskeletal microarchitectures were found to be characteristic of lens epithelial cells in humans of all ages [75] and many other species [76], and as well of trabecular meshwork cells [73]. The geodomes in lens epithelial cells appear to be a “permanent” characteristic structure rather than a transitory cytoskeletal arrangement during the cultivation of primary cells. In the case of trabecular meshwork cells, the authors suggested that the appearance of geodomes implicates the direct involvement of glucocorticoids in the pathogenesis of corticosteroid-induced glaucoma.

The geodesic domes have already been involved in the description of computational cytoskeleton models [77,78,79]. Interestingly, in some earlier studies it has also been presumed that the connecting struts of geodomes are precursors of stress fibres [63]. This assumption was elegantly confirmed by theoretical studies describing the tensegrity models of the actin cytoskeleton undergoing “stress fiber” formation [80]. Therefore, our proposed schematic model illustrated in Figure 7 is totally in line with these works. Nevertheless, by biochemical and microenvironmental modulation of the cytoskeleton reorganization, Entcheva and Bien have demonstrated that the geodesic domes can behave as rather dynamic cytoskeletal conformation controlled by external and internal forces [74]. Thus, the mechanism and functional impact of geodome formation still remains unknown.

## 5. Conclusions and Further Perspectives

Taken together, this work demonstrates that AFM imaging of single human RPE cells is feasible and can provide biophysical information on the morphological changes caused by the EMT of the cells. We found AFM-based morphological criteria such as the cell surface architecture that may allow for distinguishing single RPE cells at different stages of the dedifferentiation. 

From a future perspective, these criteria may become of clinical relevance. The EMT of RPE cells is one of the key events in the pathogenesis of PVR. Despite a marked improvement in vitreoretinal surgery over the recent years, PVR remains the main cause of treatment failure in retinal detachment surgery. Once the fibrocellular epiretinal membranes are established, PVR is difficult to cure and may result in the permanent loss of vision. Hence, there is clear evidence that patients do benefit from an early surgery, but we are still unable to predict the risk of PVR, which increases with the stage of EMT the cells present in the vitreous have undergone. Irrespective of whether aggressive pharmacological adjuvant treatment regimens or early vitreoretinal surgery should be employed in order to prevent the progression of the disease, all treatment options, and also the decision not to treat, bear considerable risks for the patient. Thus, the identification of reliable prognostic markers for PVR is a prerequisite which has not been accomplished to date. 

As shown here, AFM allows characterizing and distinguishing single RPE cells at various stages of EMT. We do believe that the results obtained from this work will serve as the solid base for the generation of a new principle for a single-cell based diagnosis of early PVR by—even more importantly—a label-free technique. In our further investigations, we will attempt to establish the correlation between the present in vitro findings with results, which will be gathered from cells isolated from the vitreous of patients suffering from PVR and related proliferative vitreoretinal diseases.

## Figures and Tables

**Figure 1 life-10-00128-f001:**
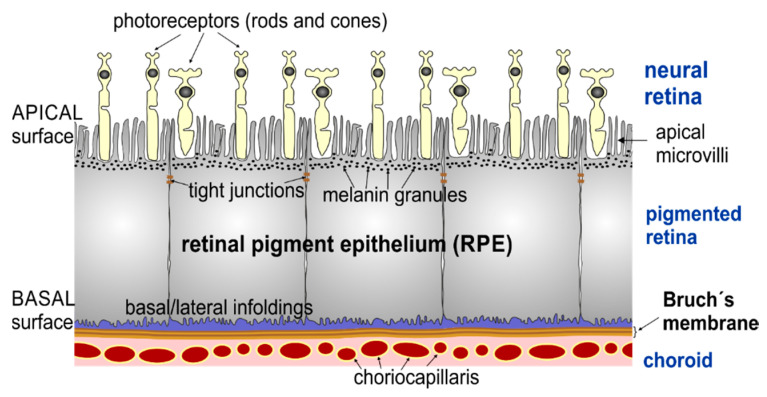
Schematic drawing of the retinal pigment epithelium and neighbouring retinal layers. The apical side of retinal pigment epithelium (RPE) cells contains different in length (from 3 up to 19 µm) actin-based microvilli (MV) [2,3,4], which are in contact with the photoreceptors (rods and cones). The basal side of the RPE cells resides on Bruch’s membrane, a multi-layer basement membrane separating the RPE from the vascularized choroid.

**Figure 2 life-10-00128-f002:**
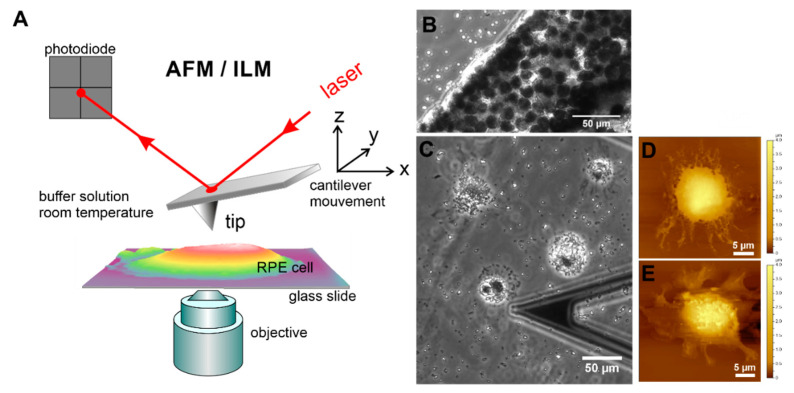
Topographical and optical imaging of RPE cells in vitro using the combination of AFM with an inverted light microscope (ILM). (**A**) Schematic drawing of AFM with optical microscopy. (**B**) Optical (phase contrast) images of native RPE membrane sheet immobilized on a glass slide. Optical (**C**) and AFM topographical images (**D**,**E**) of single suspended RPE cells from p0 on 3rd day of growth in culture. The end part of a triangular cantilever “C” is also demonstrated in optical image (**C**).

**Figure 3 life-10-00128-f003:**
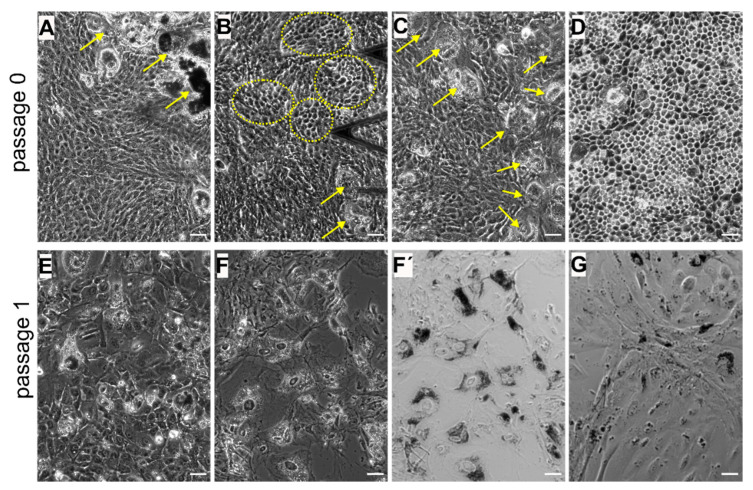
Heterogeneity of the cultured human retinal pigment epithelial (hRPE) cells at primary passages (p0 and p1). (**A**–**D**) images correspond to passage 0 and (**E**–**G**) to passage 1. (**A**,**C**) Flat epithelioids with embedded macrophage-like RPE cells (yellow arrows). (**B**) Cobblestone patterns in the milieu of flat epithelioids. (**D**) A carpet of cobblestone polygonal RPE cells. (**E**) Flat epithelioids with an enlarged size. Phase contrast (**F**) and differential interference contract (DIC) (**F′**,**G**) images illustrating large and heavily pigmented macrophage-like RPE cells. (**G**) Colony of elongated RPE cells with a fusiform morphology (fibroblast-like cells) with solitary macrophage-like cells. (**F**–**G**) images were collected on the same sample. Scale bars in all the images are 50 µm.

**Figure 4 life-10-00128-f004:**
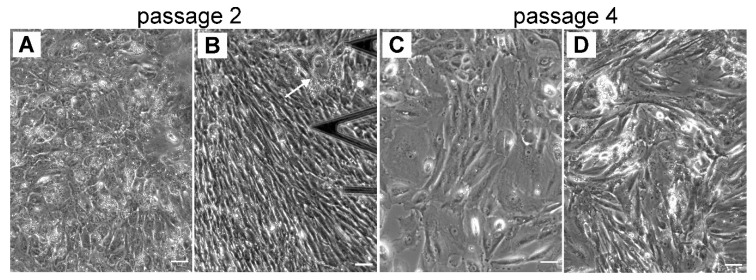
The heterogeneity of cultured hRPE cells in size and shape is increasing with in further subcultures (p2 and p4). (**A**–**B**) Typical phase contrast images for p2 and (**C**–**D**) for p4, respectively. (**A**) Illustrates flat epithelioids, (**B**) and (**D**) illustrate fibroblast-like cells. (**C**) Mixture of flat epithelioids and fibroblast-like cells. Scale bars are 50 µm.

**Figure 5 life-10-00128-f005:**
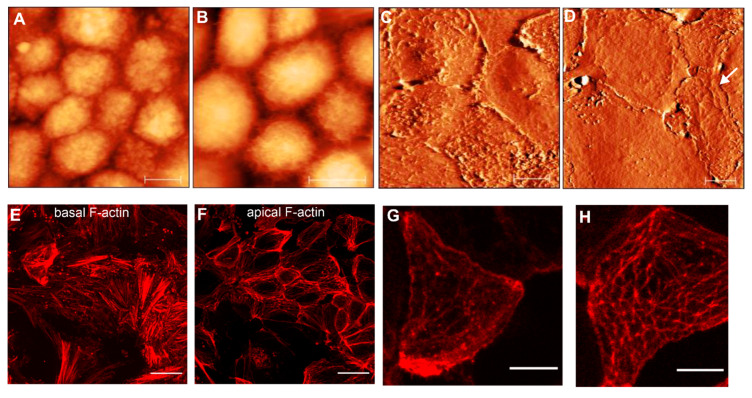
AFM topographical and immunofluorescence images of RPE epithelioids at early passages (p0 and p1). (**A**,**B**) show AFM topographical (height) images of hRPE cells with a cobblestone appearance. Color z-scale in both images is from 0 to 4.5 µm. (**C**,**D**) illustrate the AFM deflection images of cellular carpets formed by flat epithelioids. Scale bars in (**A**–**D**) are 10 µm. (**E**,**F**) images show typical F-actin arrangements at basal (**E**) and apical cell surface (**F**) in flat epithelioids. (**G**,**H**) show the high magnification of the cortical F-actin cytoskeleton in single flat epithelioids. Scale bares in the (**E**,**F**) and (**G**,**H**) images are 25 µm and 10 µm, respectively.

**Figure 6 life-10-00128-f006:**
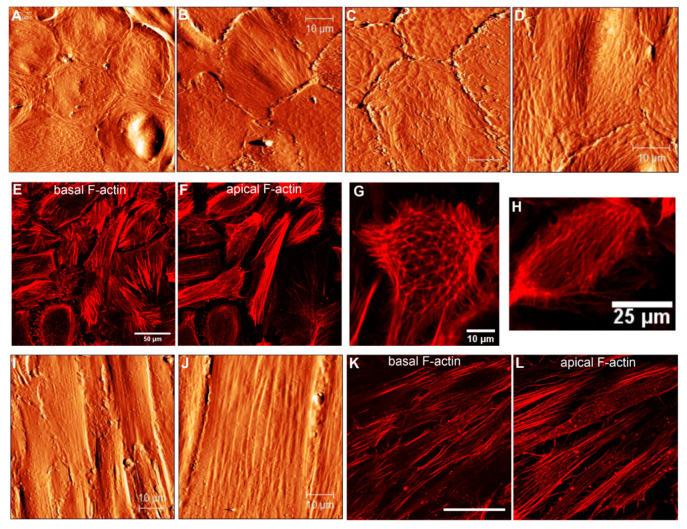
Cytoskeletal changes in cultured hRPE cells at p2 and p4. (**A**–**D**) AFM topographical images illustrating the RPE carpets of flat epithelioids. (**E**,**F**) Immunofluorescence images showing basal (**E**) and apical (**F**) F-actin arrangements in the flat epithelioids. (**G**,**H**) High-magnification fluorescent images illustrating single RPE cells with a typical geodome F-actin organization (**G**) and with parallel running F-actin filaments at the apical side. (**I**,**J**) AFM topographical images of the RPE fibroblast-like cells. Basal (**K**) and apical (**L**) F-actin cytoskeleton organization in the RPE fibroblast-like cells. Scale bars in (**A**–**D**,**I**,**J**) are 10 µm, in K it is 50 µm.

**Figure 7 life-10-00128-f007:**
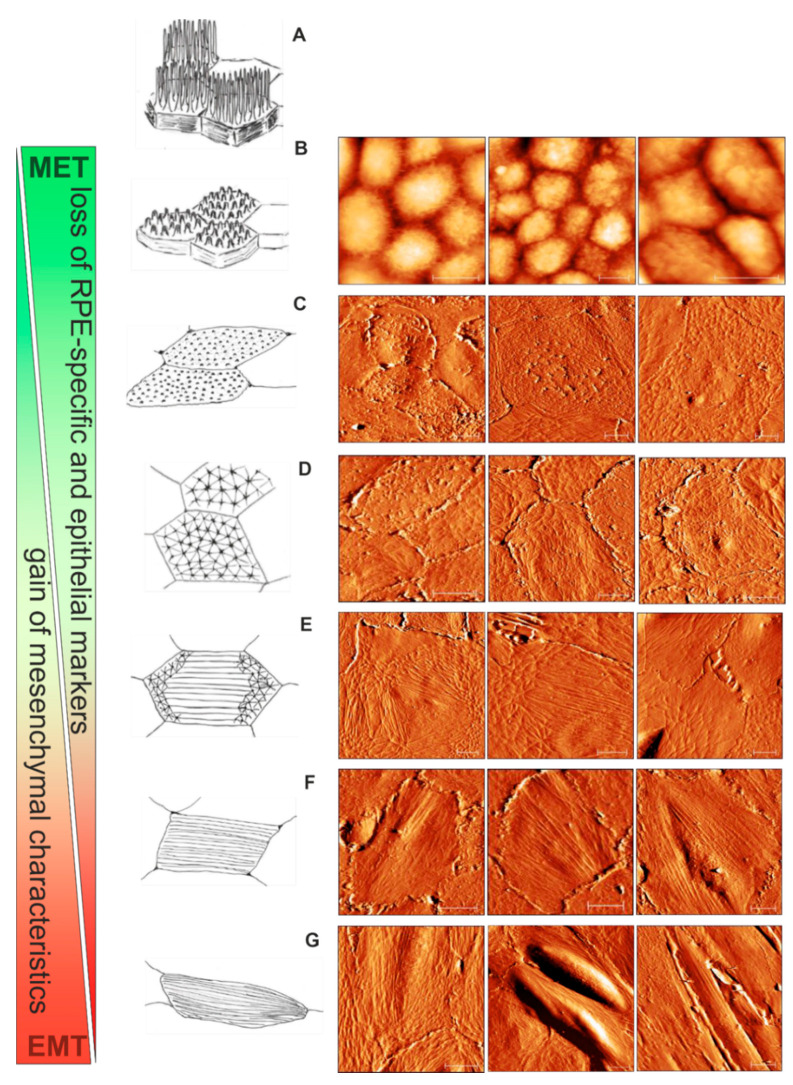
Apical cytoskeletal reorganization during the advancement of epithelial to mesenchymal transition (EMT) of hRPE cells in vitro: from cobblestone through geodome to a fibroblast-like appearance. (**A**) A schematic drawing of mature “healthy” functional hRPE cells in situ. (**B**) The cobblestone organization of the hRPE cells with retracted microvilli. (**C**) The RPE flat epithelioids with ruffles and rod-like structures. (**D**) The flat epithelioids with a distinctive geodesic dome cytoskeleton architecture. (**E**) The geodomes appear on the periphery of the flat epithelioids. (**F**) The flat epithelioids show stress fibres running along the cell body. (**G**) Appearance of RPE elongated fibroblast-like cells. Scale bars in all the AFM images are 10 µm.

**Table 1 life-10-00128-t001:** Primers used for the real-time qRT-PCR amplification.

Gene	Accession No.	Primer Sequence (Forward and Reverse)	Product Size
**ACTA2**	NM_001141945.2	5′-CTGAAGTACCCGATAGAACATGG-3′ 5′-TTGTAGAAAGAGTGGTGCCAGAT-3′	77 bp
**CDH1**	NM_004360.4	5′-CCCGGGACAACGTTTATTAC-3′ 5′-GCTGGCTCAAGTCAAAGTCC-3′	71 bp
**RPE65**	NM_000329.2	5′-CCCTCCTGCACAAGTTTGAC-3′ 5′-TCAGTCATTGCCCGTACGTA-3′	91 bp
**GNB2L**	NM_006098	5′-CTACAATGATCTTTCCCTCTAAATCC-3′ 5′-CCTAACCGCTACTGGCTGTG-3′	72 bp

**Table 2 life-10-00128-t002:** X-fold mRNA expression levels of specific RPE differentiation markers in cultured hRPE cells with different phenotypes. Error is indicated as the standard error of the mean (SEM).

Molecular Markers	Epithelioids p0	Flat Epithelioids p2 and p3	Fibroblast-Like Cells p6
RPE65	1.00 ± 0.18	0.93 ± 0.23	0.00
a-SMA	1.00 ± 0.04	2.94 ± 0.74	3.91 ± 0.85
E-cadherin	1.00 ± 0.34	0.68 ± 0.31	0.02 ± 0.02

**Table 3 life-10-00128-t003:** Roughness data corresponding to the different phenotypes with specific membrane skeleton features. The Rrms values were measured on cell areas with the fixed size of 8 × 8 µm^2^.

Cell Phenotype	Membrane Characteristics	Rrms, nm	SD, nm	Number of Cells
Cuboidal cells in monolayer	microvillous structure	333.0	95.3	6
Cuboidal cells in clusters	microvillous structure			
cluster 1		245.8	41.5	7
cluster 2		266.0	99.1	4
Flat epithelioids	ruffles and rod-like structures	85.0	30.9	7
Flat epithelioids	geodomes	43.5	6.2	9
Fibroblast-like cells	stress fibers	50.2	21.2	4

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
