# Peer review of "Nanoscopic Approach to Study the Early Stages of Epithelial to Mesenchymal Transition (EMT) of Human Retinal Pigment Epithelial (RPE) Cells In Vitro"

_life, 2020, doi:10.3390/life10080128_

Round 1
Reviewer 1 Report
The manuscript presented a report on the observation of the transformation of cultured these retinal pigment epithelial (RPE) cells through passaging. The authors observed the cells change in size, shape, but most interestingly the change in the cytoskeletal structure using AFM. This overall idea should be mentioned a lot earlier than until Figure 7 of the paper where it clearly shows the cytoskeletal changes. Another issue I have is Figure 5. They reported an observation of “geodomes” in Figure 5D, which I don’t really see it. They then state that these findings are validated by Figure 5H, which I can clearly see the geodome that they are referencing. However, the scale bar seems to be similar sizes and it is not obvious that they are focused on the same area for which they are pointing in figure 5D. Additionally they used hand sketches for figure 7 which is just a little tacky. A list of detailed comments is given in the attached pdf.

Reviewer 2 Report
Title: Nanoscopic approach to study early stages of epithelial to mesenchymal transition (EMT) of human retinal pigment epithelial (RPE) cells in vitro.
Authors: Chtechglova et al.
Manuscript ID: Life-868147
Overall Summary The authors presented a detailed and well written explanation on retinal anatomy at the cellular level which converges and focused towards the retinal pigment epithelium (RPE). The introduction went on to describe dedifferentiation of these RPE in terms of disease and damage and the on-going search for molecular cues regulating their dedifferentiation to fix or reverse the process for retinal disease treatment. While some molecular targets have been found, through signaling pathway studies, there is currently no effective pharmacological agent established and hence, more research have to be undertaken. There is further elaboration on the need to investigate RPE epithelial to mesenchymal transition (EMT), and the authors propose the use of atomic force microscopy (AFM) as a tool for studying RPE EMT, to overcome limitations of optical and electron microscopy. Donor human cells were obtained and imaged using AFM and compared against phalloidin stained samples to assess actin cytoskeletal arrangements. They qualitatively described their findings and laid claim the successful use of AFM to study RPE EMT.
While this approach does indeed overcome the limitations of current methods mentioned. Several points did arise that requires clarification/amendments/revision and should be addressed for further evaluation of the manuscript for publication in LIFE.
- The introduction, while well written, needs several points to be clarified. Apparently vitreous biopsy is possible, it should be clarified if RPE can be obtained from these biopsies and if these RPE are healthy and can be used for RPE EMT research. Is this a common approach? This gives further significance of using or proposing AFM as a technique.
- While pigmentation would impose on transmission light microscopy. This reviewer is unsure if it will hinder immunofluorescence microscopy, especially if actin cytoskeleton is of interest. (line 93-94) Could the authors please expand on this, or use a reference to substantiate this sentence?
- The authors also mentioned that (line 98-100) AFM have the possibility of examining biological processes under/or near physiological conditions without sample preparation (labeling or coating). Could the authors further clarify or report in results, the time scale of their AFM measurements/imaging as compared to biological processes that is of interest in the RPE EMT field?
- The biggest drawback of this manuscript is the reliance on qualitative descriptive to ‘analyze’ the AFM images. Evaluation of histological images by pathologies are phasing towards automation of results, even using machine learning, to eliminate false positives or false negatives. The authors need to be able to quantify, from AFM images, differences in the structures, to push AFM as a standard tool in this field. Presumably this is the aim, looking at the first author’s expertise. In Figure 7, they nicely sketched schematics to match their AFM images. Suggestion is to binarize, or any other image processing workflow(s), several AFM images to show that these schematics are representative of at least 6-8 RPE cells at each stage of EMT advancement. This can expand section 3.2.
- Please also discuss, or perform, the possible use of automatically quantifying binarized images from AFM using algorithms to further classify these structural morphologies. Alhussein et al., developed an approach that could quantify cytoskeleton arrangement in a temporal manner. https://doi.org/10.1002/cm.21297
- Figure 5. Is the bottom row a reflection of the cytoskeleton arrangement for the cells in the top row? If yes, the unique structure of Figure 5H does not seem to absent from Figure 5D.
Round 2
Reviewer 2 Report
I am satisfied with the answers to my queries.